# Application of recommended preventive measures against COVID-19 could help mitigate the risk of SARS-CoV-2 infection during dental practice: *Results from a follow-up survey of French dentists*

Hadrien Diakonoff[1,2☉], Sébastien Jungo[3☉], Nathan Moreau[3,4], Marco E. Mazevet[5], Anne-Laure Ejeil[3,6], Benjamin Salmon[3,6‡], Violaine Smaïl-Faugeron[3,7‡*]

1 Dental Medicine Department, AP-HP, Mondor Hospital, Créteil, France, 2 Inserm UMR S 1145, Institut Droit et Santé, Université de Paris, Paris, France, 3 Dental Medicine Department, AP-HP, Bretonneau Hospital, Paris, France, 4 Université de Paris, Laboratory of Orofacial Neurobiology (EA 7543), Paris, France, 5 Dental Innovation and Translation Hub, Faculty of Dentistry, Oral & Craniofacial Sciences, Kings College London, Guy's Hospital, London, United Kingdom, 6 Université de Paris, Laboratory of Orofacial Pathologies, Imaging and Biotherapies, Montrouge, France, 7 Université de Paris, EA 7323 Pharmacologie Et Thérapeutique de L'enfant Et de La Femme Enceinte, Paris, France

☉ These authors contributed equally to this work.
‡ These authors are co-senior authors on this work.
* violaine.smail-faugeron@u-paris.fr

## Abstract

### Background

During the first-wave of the COVID-19 pandemic, dentists were considered at high-risk of infection. In France, to stop the spread of SARS-CoV-2, a nationwide lockdown was enforced, during which dentists suspended their routine clinical activities, working solely on dental emergencies. This measure has had an indisputable mitigating effect on the pandemic. To continue protecting dentists after suspension of nationwide lockdown, implementation of preventive measures was recommended, including adequate personal protective equipment (PPE) and room aeration between patients. No study has explored whether implementation of such preventive measures since the end of the first-wave has had an impact on the contamination of dentists.

### Methods

An online survey was conducted within a French dentist population between July and September 2020. To explore risk factors associated with COVID-19, univariate and multivariate logistic regression analyses were performed.

### Results

The results showed that COVID-19 prevalence among the 3497 respondents was 3.6%. Wearing surgical masks during non-aerosol generating procedures was a risk factor of COVID-19, whereas reducing the number of patients was a protective factor.

**Data Availability Statement:** All relevant data are within the manuscript and its Supporting information files.

**Funding:** The author(s) received no specific funding for this work.

**Competing interests:** The authors have declared that no competing interests exist.

## Conclusions

Considering the similar COVID-19 prevalence between dentists and the general population, such data suggest that dentists are not overexposed in their work environment when adequate preventive measures are applied.

## Impact

Dentists should wear specific PPE (FFP2, FFP3 or (K)N95 masks) including during non-aerosol generating procedures and reduce the number of patients to allow proper implementation of disinfection and aeration procedures. Considering the similarities between COVID-19 and other viral respiratory infections, such preventive measures may also be of interest to limit emerging variants spread as well as seasonal viral outbreaks.

## Introduction

On March 11, 2020, the World Health Organization declared Coronavirus disease 2019 (COVID-19), caused by a novel severe acute respiratory syndrome coronavirus 2 (SARS-CoV-2), a global pandemic [1]. In France, to help stop the spread of the virus, a nationwide lockdown was enforced by the government on March 17, 2020 [2]. At that time, healthcare workers were considered at high-risk of infection, especially dentists [3, 4]. Thus, during lockdown, private practices have suspended their routine clinical activities to form an emergency only dental service, with hospital units remaining open for the same urgent treatments [5]. In a previous study including 4172 French dentists surveyed in April 2020, changes in work rhythm or clinical practice (e.g., participation in telephone regulation of emergency cases and / or practice limited to emergencies only) following lockdown appeared to be protective factors against COVID-19, whereas working in dental specialties highly exposed to droplets such as periodontology might be an at-risk practice [6]. However, very few people had been tested at that time (<5%), namely only symptomatic people or those with risk factors for severe COVID-19, in adherence with French government policy of the time [7]. After the suspension of lockdown on May 11, 2020, testing policy changed providing easier access to testing for healthcare workers, including reverse transcription–quantitative polymerase chain reaction (RT-qPCR) and serology tests [8]. Moreover, preventive measures were recommended such as the reinforcement of disinfection procedures between patients and implementation of specific personal protective equipment (PPE), in particular FFP2 masks during aerosol generating procedures [9].

As a logical continuation of our previous study, this study aimed to resurvey French dentists after the first French lockdown (1) to report the prevalence of COVID-19, (2) to assess the impact of preventive measures implemented following the end of the lockdown, and (3) to identify risk indicators associated with COVID-19.

## Methods

From July 8 to September 8, 2020, an anonymous, non-incentivized, online survey was conducted in accordance with the 1964 Helsinki declaration and approved by the French national authorities regulating confidentiality (CNIL, Commission Nationale Informatique et Libertés, No. 2217408). Participants were informed of the data collection, study aims and relevant data protection measures. Survey setting was equivalent to the first questionnaire sent in April [6].

## Survey development

In total, 32 questions were divided in 8 sections, with a mean number of questions per section of 4 (see S1 Fig). The questionnaire consisted of several categories: sociodemographic data (gender, age); factors associated with COVID-19-related death [10]; perceived stress relating to the COVID-19 pandemic during the lockdown and after its suspension; work environment before the pandemic and after the suspension of lockdown; and actual COVID-19 status. Perceived stress levels of respondents were assessed with a numerical rating scale (NRS) ranging from 0 (no stress) to 10 (highest stress imaginable) [11], regarding their personal safety, the safety of their families and patients, and the financial stability of their professional practice. Usual work environment characteristics (i.e. before the enforcement of lockdown on March 17, 2020) included the use of public transportation, type of practice (dental office and/or hospital department) and professional orientation (general practice or dental specialty). Work environment characteristics after suspension of lockdown included use of public transportation and professional exposure (i.e. number of daily treated patients, number of aerosol vs. non-aerosol dental procedures, and types of PPE used). COVID-19 status included laboratory test for COVID-19 performed (RT-qPCR test by nasopharyngeal swab or serology test) and self-reported symptoms.

## Data synthesis and analysis

Binary variables were described using frequencies (percentages) and continuous variables were described using median (interquartile range (IQR)). When appropriate, Chi-squared or Fisher's exact test were used for binary variables and Kruskal-Wallis for continuous variables to compare differences between SARS-CoV-2 positive vs. SARS-CoV-2 negative or non-tested cases. To explore the associated risk indicators, univariate and multivariate logistic regression analyses were performed. Variables with p value $\leq$ 0.2 in the univariate analysis were introduced into the multivariate analysis. Then, covariate selection was done with a stepwise descending procedure based on Akaike Information Criteria. The false discovery rate was controlled at a level of 5% with a Benjamini and Hochberg procedure [12]. A random region effect was then introduced to account for local disparities. Analyses involved use of R (version 4.0.3; www.r-project.org).

# Results

In total, 3497 dentists responded to the questionnaire, which corresponds to approximately 9% of French dentists. Half of them responded to the first survey (1886, 53.9%).

## Socio-demographic data, medical conditions, and clinical practice before the pandemic

The median age of respondents was 53 years (IQR, 42 to 61), ranging from 24 to 79 years, and more than half were women. About one fifth of respondents (19.8%, n = 695) had one or more risk factors for critical and mortal COVID-19 cases, of which the most common were being overweight or obese, tobacco consumption, hypertension, cancer, cardiovascular and chronic obstructive pulmonary diseases. Most dentists worked in private practices (3415 [97.7%]). General practice was the most represented practice (3118 [82.2%]), followed by orthodontics and practice limited to oral surgery or periodontology. Details are listed in Table 1.

## Prevalence of COVID-19

From January to September 2020, 3.6% of respondents (n = 126) were tested positive for COVID-19. Among those, 13 (10.3%) were confirmed by RT-qPCR test only, 68 (54%) by

**Table 1. Socio-demographic data, medical conditions, and clinical practice before the pandemic.**

| | All included dentists (n = 3497) | No test performed (n = 2476) | Tested Negative (n = 895) | Tested Positive (n = 126) | p-value |
|---|---|---|---|---|---|
| **Demographic data** | | | | | |
| Age, years | 53 [42, 61] | 53 [42, 61] | 54 [42, 61] | 54 [41.25, 61] | 0.698* |
| Female gender | 1847 (52.8) | 1277 (51.6) | 508 (56.8) | 62 (49.2) | **0.02** |
| **Medical Conditions** | | | | | |
| Current pregnancy | 47 (1.3) | 34 (1.4) | 10 (1.1) | 3 (2.4) | 0.426 |
| Current Smoking | 270 (7.7) | 192 (7.8) | 66 (7.4) | 12 (9.5) | 0.638 |
| Comorbidities | | | | | |
| Allergies | 463 (13.2) | 320 (12.9) | 128 (14.3) | 15 (11.9) | 0.544 |
| Diabetes | 67 (1.9) | 38 (1.5) | 21 (2.3) | 8 (6.3) | **0.002** |
| Hypertension | 284 (8.1) | 184 (7.4) | 88 (9.8) | 12 (9.5) | 0.062 |
| Cardiopathies | 109 (3.1) | 76 (3.1) | 30 (3.4) | 3 (2.4) | 0.86 |
| COPD | 97 (2.8) | 68 (2.7) | 26 (2.9) | 3 (2.4) | 0.959 |
| CKD | 19 (0.5) | 11 (0.4) | 8 (0.9) | 0 (0.0) | 0.276 |
| Malignancies | 114 (3.3) | 76 (3.1) | 34 (3.8) | 4 (3.2) | 0.554 |
| Overweight or obesity | 339 (9.7) | 223 (9.0) | 96 (10.7) | 20 (15.9) | **0.023** |
| ID | 41 (1.2) | 28 (1.1) | 11 (1.2) | 2 (1.6) | 0.782 |
| Other | 98 (2.8) | 71 (2.9) | 26 (2.9) | 1 (0.8) | 0.444 |
| **Clinical practice** | | | | | |
| Structure | | | | | **<0.001** |
| Private practice | 3295 (94.3) | 2371 (95.8) | 811 (90.6) | 113 (89.7) | |
| Hospital | 70 (2.0) | 26 (1.1) | 37 (4.1) | 7 (5.6) | |
| Private practice and hospital | 120 (3.4) | 69 (2.8) | 46 (5.1) | 5 (4.0) | |
| Other | 10 (0.3) | 8 (0.3) | 1 (0.1) | 1 (0.8) | |
| Practice | | | | | **<0.001#** |
| General practice | 3118 (89.2) | 2235 (90.3) | 781 (87.3) | 102 (81.0) | |
| Specialized practice | 171 4.9) | 104 (4.2) | 53 (5.9) | 14 (11.1) | |
| Orthodontics | 185 (5.3) | 125 (5.1) | 51 (5.7) | 9 (7.1) | |
| Other | 21 (0.6) | 10 (0.4) | 10 (1.1) | 1 (0.8) | |
| Specific specialty | | | | | |
| Endodontics | 26 (0.7) | 12 (0.5) | 11 (1.2) | 3 (2.4) | **0.008** |
| Oral surgery | 55 (1.6) | 36 (1.5) | 17 (1.9) | 2 (1.6) | 0.597 |
| Pediatric dentistry | 42 (1.2) | 24 (1.0) | 14 (1.6) | 4 (3.2) | **0.042** |
| Periodontology | 57 (1.6) | 41 (1.7) | 12 (1.3) | 4 (3.2) | 0.276 |

Data are median [IQR], n (%). P-values comparing dentists' COVID-19 test status (no test, negative or positive) are from (#) Chi-Square, (*) Kruskal-Wallis or Fisher's exact test when not specified. COPD: chronic obstructive pulmonary disease; CKD: chronic kidney disease; ID: immunodeficiencies.

serology test only, and 45 (35.7%) by both tests. In total, 1021 (28.3%) respondents were tested, including 198 (20%) with RT-qPCR test, 651 (63.8%) with serology test and 172 (16.8%) with both tests. Half of tested respondents (n = 511) reported at least one COVID-compatible symptom. Among the 126 COVID-19 positive cases, 108 (85.7%) were symptomatic, and the most common symptoms were tiredness, fever, anosmia, cough, headache, and ageusia. Moreover, 41 (32.5%) suspected a transmission within their work environment and 33 (26.2%) within the private sphere. Details are listed in Table 2.

Only 3 (2.4%) cases may have been infected after May 11, 2020; most of cases (108, 85.7%) may have been infected before this date, and data were insufficient to assess the date of infection for 15 (11.9%) cases (Fig 1). In addition, the peak of COVID-19 infection for this dentist

**Table 2. Symptoms and putative exposure history in dentists.**

| | All included dentists (n = 3497) | No test performed (n = 2476) | Tested Negative (n = 895) | Tested Positive (n = 126) | p-value |
|---|---|---|---|---|---|
| **Symptoms** | | | | | |
| None | 2671 (76.4) | 2161 (87.3) | 492 (55.0) | 18 (14.3) | <0.001 |
| Fever (>38˚) | 301 (8.6) | 90 (3.6) | 148 (16.5) | 63 (50.0) | <0.001 |
| Chills | 185 (5.3) | 55 (2.2) | 93 (10.4) | 37 (29.4) | <0.001 |
| Headache | 368 (10.5) | 140 (5.7) | 173 (19.3) | 55 (43.7) | <0.001 |
| Conjunctivitis | 53 (1.5) | 21 (0.8) | 22 (2.5) | 10 (7.9) | <0.001 |
| Tiredness | 573 (16.4) | 213 (8.6) | 274 (30.6) | 86 (68.3) | <0.001 |
| Rhinitis | 223 (6.4) | 85 (3.4) | 111 (12.4) | 27 (21.4) | <0.001 |
| Myalgia | 270 (7.7) | 83 (3.4) | 132 (14.7) | 55 (43.7) | <0.001 |
| Sore throat | 254 (7.3) | 94 (3.8) | 134 (15.0) | 26 (20.6) | <0.001 |
| Cough | 344 (9.8) | 108 (4.4) | 180 (20.1) | 56 (44.4) | <0.001 |
| Anosmia | 120 (3.4) | 26 (1.1) | 37 (4.1) | 57 (45.2) | <0.001 |
| Ageusia | 114 (3.3) | 28 (1.1) | 33 (3.7) | 53 (42.1) | <0.001 |
| Dyspnea | 114 (3.3) | 36 (1.5) | 48 (5.4) | 30 (23.8) | <0.001 |
| ARDS | 21 (0.6) | 4 (0.2) | 11 (1.2) | 6 (4.8) | <0.001 |
| Dizziness | 89 (2.5) | 27 (1.1) | 50 (5.6) | 12 (9.5) | <0.001 |
| Other | 121 (3.5) | 32 (1.3) | 68 (7.6) | 21 (16.7) | <0.001 |
| **Contact history** | | | | | |
| Does not believe to be infected | 2839 (81.2) | 2195 (88.7) | 629 (70.3) | 15 11.9) | <0.001 |
| Unknown | 316 (9.0) | 159 (6.4) | 127 (14.2) | 30 (23.8) | <0.001 |
| Dental procedures | 180 (5.2) | 65 (2.6) | 70 (7.8) | 45 (35.7) | <0.001 |
| Spouse, child | 85 (2.4) | 39 (1.6) | 26 (2.9) | 20 (15.9) | <0.001 |
| During public transportation or travel | 44 (1.3) | 9 (0.4) | 22 (2.5) | 13 (10.3) | <0.001 |
| Coworker | 32 (0.9) | 12 (0.5) | 12 (1.3) | 8 (6.3) | <0.001 |
| Assistant, secretary | 22 (0.6) | 11 (0.4) | 8 (0.9) | 3 (2.4) | 0.021 |
| Other | 95 (2.7) | 51 (2.1) | 32 (3.6) | 12 (9.5) | <0.001 |

Data are n (%). P-values comparing dentists' COVID-19 test status (no test, negative or positive) are from Fisher's exact test. ARDS: acute respiratory distress syndrome.

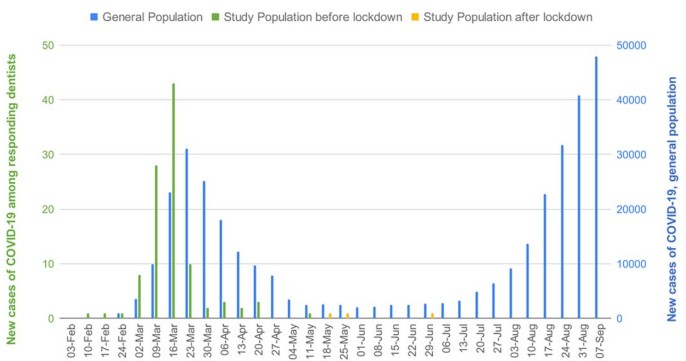

**Fig 1. Weekly evolution of new cases of Covid-19 in France.**

sample (around March 16, 2020) appeared earlier than for the general population (around March 23, 2020).

## Implementation of preventive measures following suspension of nationwide lockdown

After suspension of lockdown, most respondents (97.9%, n = 3424) returned to work. The use of public transportation was reduced by 40.9% (181 [5.3%] before the pandemic vs 107 [3.1%] after suspension of lockdown). Most respondents reduced the number of patients treated (77.1%, n = 2694) and the number of dental procedures (27.3%, n = 955). More participants wore FFP2, FFP3 or (K)N95 masks during aerosol generating procedures than during non-aerosol generating procedures (3294 [94.2%] vs. 2219 [63.5%]). The same trend was observed for safety goggles (3298 [94.4%] vs. 1578 [45.2%]), disposable gown (2851 [81.6%] vs. 1185 [33.9%]), hairnets (2984 [85.4%] vs. 2056 [58.8%]) and shoe covers (450 [12.9%] vs. 268 [7.7%]). Overall, dentists were more anxious regarding contaminating their families (median NRS score = 5 [IQR, 2 to 7]) and their professional financial and organizational difficulties (5 [3 to 7]) than to be contaminated or to contaminate their patients (3 [2 to 6]). Details are given in Table 3.

## Risk indicators associated with COVID-19

In the univariate analysis, odds of COVID-19 were higher in males, in dentists with specific comorbidities such as diabetes, thyroid disease and being overweight or obese, users of public transportation, in dentists working in hospital, with a specialized practice, in particular a practice limited to pediatric dentistry, and dentists who wore surgical masks and shoe covers during aerosol or non-aerosol generating procedures, whereas odds were lower in dentists treating fewer patients and wearing FFP2, FFP3 or (K)N95 masks during aerosol or non-aerosol generating procedures. In the multivariate analysis, dentists with diabetes (OR 2.83 IC 95% [1.21 to 6.61], p = 0.026), thyroid disease (4.07 [1.36 to 12.17, p = 0.024), and overweight or obese (1.83 [1.08 to 3.10], p = 0.032), users of public transportation before the lockdown (2.56 [1.46 to 4.48, p = 0.004), wearing surgical masks during non-aerosol generating procedures (1.91 [1.32 to 2.76], p = 0.004), and shoe covers during non-aerosol generating procedures (2.33 [1.09 to 5.01], p = 0.034) were associated with increased odds of COVID-19, whereas reducing the number of patients was associated with decreased odds (0.54 [0.37 to 0.80], p = 0.005). When introducing a random region effect in the multivariate analysis, odds of COVID-19 remained higher only in dentists wearing surgical masks during non-aerosol generating procedures (1.88 [1.30 to 2.73], p = 0.008), and odds remained lower in dentists treating fewer patients (0.56 [0.38 to 0.83], p = 0.016). Details are given in Table 4.

## Discussion

This large survey followed a previous study assessing prevalence and risk indicators of first-wave COVID-19 among French dentists. To our best knowledge, this second study is the first to assess whether clinical practices have changed since the end of the first-wave pandemic, with specific focus on the putative impact of implementation of preventive measures.

At the time of data collection (September 8, 2020), our results confirmed that there was no strong evidence to confirm that dentists were at higher risk of COVID-19 than the general population (3.6% of dentists vs. 5.2% of the general population, www.santepubliquefrance.fr), workers in hospital settings (3.4%), nor than healthcare workers (4.0%) [13]. We also found that most infections occurred before French nationwide lockdown and probably almost none after the suspension of lockdown. This could be explained by (1) the global decline in

**Table 3. Clinical practice and perceived stress after the lifting of the lockdown.**

| | All included dentists (n = 3497) | No test performed (n = 2476) | Tested Negative (n = 895) | Tested Positive (n = 126) | p-value |
|---|---|---|---|---|---|
| **Return to work** | | | | | 0.079 |
| Yes | 3424 (97.9) | 2427 (98.0) | 875 (97.8) | 122 (96.8) | |
| Telephone regulation | 8 (0.2) | 5 (0.2) | 3 (0.3) | 0 (0.0) | |
| No | 44 (1.2) | 25 (1.0) | 15 (1.7) | 4 (3.2) | |
| Retired | 21 (0.6) | 19 (0.8) | 2 (0.2) | 0 (0.0) | |
| **Taking public transportation** | | | | | |
| Before lockdown | 181 (5.3) | 95 (3.9) | 70 (8.0) | 16 (13.1) | **<0.001** |
| After lockdown | 107 (3.1) | 59 (2.4) | 36 (4.1) | 12 (9.8) | **<0.001** |
| **Changes after lockdown** | 270 (7.7) | 83 (3.4) | 132 (14.7) | 55 (43.7) | **<0.001** |
| No change | 941 (26.9) | 658 (26.6) | 241 (26.9) | 42 (33.3) | 0.249 |
| Reducing number of patients | 2694 (77.1) | 1921 (77.6) | 692 (77.3) | 81 (64.3) | **0.004** |
| Reducing number of dental procedures | 955 (27.3) | 698 (28.2) | 233 (26.0) | 24 (19.0) | **0.046** |
| Reduce number of medical staff | 120 (3.4) | 84 (3.4) | 30 (3.4) | 6 (4.8) | 0.653 |
| Reduce number of paramedical staff | 184 (5.3) | 127 (5.1) | 50 (5.6) | 7 (5.6) | 0.825 |
| Treating emergencies only | 31 (0.9) | 22 (0.9) | 8 (0.9) | 1 (0.8) | 1 |
| Other | 120 (3.4) | 78 (3.2) | 36 (4.0) | 6 (4.8) | 0.285 |
| **PPE (aerosol generating procedures)** | | | | | |
| Surgical mask | 699 (20.0) | 474 (19.2) | 188 (21.0) | 37 (29.4) | **0.017** |
| FFP2/FFP3/(K)N95 mask | 3294 (94.2) | 2331 (94.2) | 851 (95.1) | 112 (88.9) | **0.029** |
| Safety goggles | 3298 (94.4) | 2337 (94.5) | 846 (94.5) | 115 (91.3) | 0.31 |
| Hairnets | 2984 (85.4) | 2094 (84.6) | 782 (87.4) | 108 (85.7) | 0.137 |
| Shoe covers | 450 (12.9) | 301 (12.2) | 124 (13.9) | 25 (19.8) | **0.029** |
| Disposable gown | 2848 (81.5) | 2027 (81.9) | 724 (80.9) | 97 (77.0) | 0.317 |
| **PPE (non-aerosol generating procedures)** | | | | | |
| Surgical mask | 1307 (37.4) | 881 (35.6) | 360 (40.2) | 66 (52.4) | **<0.001** |
| FFP2/FFP3/(K)N95 mask | 2219 (63.5) | 1601 (64.7) | 558 (62.3) | 60 (47.6) | **<0.001** |
| Safety goggles | 1578 (45.2) | 1106 (44.7) | 406 (45.4) | 66 (52.4) | 0.237 |
| Hairnets | 2056 (58.8) | 1430 (57.8) | 555 (62.0) | 71 (56.3) | 0.074 |
| Shoe covers | 268 (7.7) | 183 (7.4) | 66 (7.4) | 19 (15.1) | **0.013** |
| Disposable gown | 1185 (33.9) | 843 (34.1) | 298 (33.3) | 44 (34.9) | 0.884 |
| **Perceived stress** | | | | | |
| Global | 5 [3, 7] | 5 [3, 7] | 5 [3, 7] | 5 [3, 7] | 0.618* |
| Personal safety | 3 [1, 5] | 3 [1, 5] | 3 [1, 5] | 3 [2, 7] | <0.001* |
| Safety of their families | 5 [2, 7] | 5 [2, 7] | 5 [2, 8] | 7 [5, 8] | <0.001* |
| Safety of their patients | 2 [0, 5] | 2 [0, 5] | 2 [0, 5] | 3 [0, 5] | 0.058* |
| Professional practice | 7 [5, 8] | 7 [5, 8] | 7 [5, 8] | 7 [5, 9] | 0.315* |

Data are median [IQR], n (%). P-values comparing dentists' COVID-19 test status (no test, negative or positive) are from (*) Kruskal-Wallis or Fisher's exact test when not specified.

SARS-CoV-2 circulation [14], (2) the indisputable mitigating effect of lockdown enforcement [15], and (3) the implementation of preventive measures, including adequate specific PPE enforced after lockdown [16, 17] and room ventilation between patients [18]. In our sample, the use of PPE was massively adopted during aerosol generating procedures, such as wearing FFP2, FFP3 or (K)N95 masks or safety googles (around 94%). Moreover, three quarters of the respondents treated fewer patients, and the multivariate analysis showed that reducing the

**Table 4. Risk indicators associated with COVID-19 among dentists.**

| | No test performed or tested negative (n = 3371) | Tested Positive (n = 126) | Univariate OR (95% CI, p-value) | Multivariate OR (95% CI, p-value) | Multivariate OR (95% CI, p-value)* |
|---|---|---|---|---|---|
| **Medical Conditions** | | | | | |
| Diabetes | 59 (88.1) | 8 (11.9) | **3.80 (1.78–8.14, p = 0.001)** | **2.83 (1.21–6.61, p = 0.026)** | **2.49 (1.03–6.00, p = 0.056)** |
| Overweight or obesity | 319 (94.1) | 20 (5.9) | **1.80 (1.10–2.95, p = 0.019)** | **1.83 (1.08–3.10, p = 0.032)** | 1.78 (1.04–3.02, p = 0.054) |
| Thyroid disease | 30 (88.2) | 4 (11.8) | **3.65 (1.27–10.52, p = 0.017)** | **4.07 (1.36–12.17, p = 0.024)** | 3.85 (1.27–11.67, p = 0.045) |
| **Taking public transportation** | | | | | |
| Before lockdown | 165 (91.2) | 16 (8.8) | **2.87 (1.66–4.97, p<0.001)** | **2.56 (1.46–4.48, p = 0.004)** | 1.63 (0.88–3.02, p = 0.136) |
| After lockdown | 95 (88.8) | 12 (11.2) | **3.68 (1.96–6.91, p<0.001)** | - | - |
| **Changes after lockdown** | | | | | |
| No change | 899 (95.5) | 42 (4.5) | 1.37 (0.94–2.01, p = 0.100) | - | - |
| Reducing number of patients | 2613 (97.0) | 81 (3.0) | **0.52 (0.36–0.76, p = 0.001)** | **0.54 (0.37–0.80, p = 0.005)** | **0.56 (0.38–0.83, p = 0.016)** |
| Reducing number of dental procedures | 931 (97.5) | 24 (2.5) | **0.62 (0.39–0.97, p = 0.035)** | - | - |
| **PPE (aerosol generating procedures)** | | | | | |
| Surgical mask | 662 (94.7) | 37 (5.3) | **1.70 (1.15–2.52, p = 0.008)** | - | - |
| FFP2/FFP3/(K)N95 mask | 3182 (96.6) | 112 (3.4) | **0.47 (0.26–0.84, p = 0.010)** | - | - |
| Shoe covers | 425 (94.4) | 25 (5.6) | **1.71 (1.09–2.69, p = 0.019)** | 0.98 (0.50–1.94, p = 0.963) | 0.98 (0.50–1.95, p = 0.959) |
| Disposable gown | 13 (86.7) | 2 (13.3) | 4.16 (0.93–18.65, p = 0.062) | - | - |
| **PPE (non-aerosol generating procedures)** | | | | | |
| Surgical mask | 1241 (95.0) | 66 (5.0) | **1.89 (1.32–2.69, p<0.001)** | **1.91 (1.32–2.76, p = 0.004)** | **1.88 (1.30–2.73, p = 0.008)** |
| FFP2/FFP3/(K)N95 mask | 2159 (97.3) | 60 (2.7) | **0.51 (0.36–0.73, p<0.001)** | - | - |
| Shoe covers | 249 (92.9) | 19 (7.1) | **2.22 (1.34–3.69, p = 0.002)** | **2.33 (1.09–5.01, p = 0.034)** | 2.31 (1.07–4.98, p = 0.054) |

OR = odds ratio; 95% CI = 95% confident interval. PPE: personal protective equipment.

* Multivariate analysis with random region effect.

number of patients was a specific protective indicator against COVID-19. Indeed, treating fewer patients allows proper implementation of disinfection and ventilation procedures between patients [19, 20]. This is consistent with the results of our first study, showing that changing one's work rhythm was associated with decreased odds of COVID-19.

Although dentists were surprisingly not at higher risk of COVID-19 than the general population, we showed that the peak of infection for dentists occurred one week earlier than for the general population. This may highlight that dentists could have been overexposed to COVID-19 before the enforced lockdown and the implementation of preventive measures.

Interestingly, the multivariate analysis showed that wearing a surgical mask during non-aerosol generating procedures was a specific risk indicator of COVID-19. Some authors suggest that the practice of aerosol-generating procedures within a saliva-rich environment could be a major transmission route for respiratory viruses [18, 21, 22] whilst others have advocated that no copies of the SARS-COV-2 can be found in these aerosols, when appropriate prevention measures are taken [23]. However, during non-aerosol generating procedures, such as clinical interviewing or examination, the patient can talk, cough, scream or cry, all of which can also cause saliva projections and produce contaminated aerosols [24]. Wearing specific PPE (in particular FFP2, FFP3 or (K)N95 masks) should be warranted, including during non-aerosol generating procedures, with an emphasis on ventilation that can be indirectly monitored through the usage of $CO_2$ readers [25].

Wearing shoe covers during non-aerosol generating procedures seemed to be a risk indicator of COVID-19. Actually, this variable was strongly associated with practice limited to periodontology (p = 0.01), a confounding variable. This is consistent with the results of our first study. Not only periodontologists seem to be highly exposed to airborne droplets [26, 27], but they also spend time on clinical interviews or examinations during which they could be infected especially if they did not wear specific PPE mask or wear it incorrectly [28]. This assumption could be extended to practice limited to pediatric dentistry, which was associated with increased odds of COVID-19 in the univariate analysis. Indeed, dentists are often closer to children than adults, and there are more contacts due to children motion and behavior.

We also showed other risk indicators of COVID-19, such as specific comorbidities (diabetes, thyroid disease, being overweight or obese), in adherence with risk factors identified in previous studies [29]. Using public transportation before lockdown was also associated with increased odds of COVID-19, similarly to previous results showing an increased risk of respiratory virus transmission due to proximity in a closed environment [30]. These results thus confirmed those found in our previous study.

After having introduced a random region effect in the multivariate analysis, reducing the number of patients still remained a protective indicator against COVID-19 and only wearing surgical masks during non-aerosol generating procedures remained a specific risk indicator of COVID-19. This could suggest that the aforementioned comorbidities, use of public transportation or having a limited practice such as periodontology could actually be factors associated with densely populated areas.

Our study has several limitations. First, the prevalence of COVID-19 among dentists could have been underestimated, as only one third of respondents have been tested. Nevertheless, the number of tested respondents has increased six-fold compared to the first study (<5%) [6], thus increasing its robustness. Second, it was not possible to establish causal relationships between being tested positive for COVID-19 and wearing a surgical mask during non-aerosol generating procedures. In the univariate analysis, we showed that COVID-19 positive respondents were less stressed for their personal health and wore fewer FFP2, FFP3 or (K)N95 masks during aerosol or non-aerosol generating procedures. It cannot be excluded that the infected dentists took higher risks by using less protection. Third, it was difficult to assign a date of contamination for people tested by serology. However, we asked for the date of onset of symptoms to try to get as close as possible to said date.

In conclusion, although dentists had a similar prevalence of COVID-19 infection as compared to the general population, our results suggest that they could be overexposed to COVID-19 without the implementation of specific preventive measures. In particular, dentists should reduce the number of patients to allow proper implementation of disinfection and ventilation procedures and wear specific PPE (FFP2, FFP3 or (K)N95 masks) including during non-aerosol generating procedures. Considering the similarities between COVID-19 and other viral

respiratory infections, these preventive measures may also be applicable to limit emerging variants spread as well as seasonal viral outbreaks.

## Supporting information

**S1 Fig.**
(TIF)

**S1 Data.**
(CSV)

## Acknowledgments

We would like to thank all oral health-care workers for participating in this survey.

## Author Contributions

**Conceptualization:** Hadrien Diakonoff, Sébastien Jungo, Nathan Moreau, Marco E. Mazevet, Anne-Laure Ejeil, Benjamin Salmon, Violaine Smaïl-Faugeron.

**Data curation:** Hadrien Diakonoff, Sébastien Jungo, Violaine Smaïl-Faugeron.

**Formal analysis:** Hadrien Diakonoff, Sébastien Jungo, Violaine Smaïl-Faugeron.

**Funding acquisition:** Hadrien Diakonoff, Sébastien Jungo, Violaine Smaïl-Faugeron.

**Methodology:** Hadrien Diakonoff, Sébastien Jungo, Benjamin Salmon, Violaine Smaïl-Faugeron.

**Supervision:** Violaine Smaïl-Faugeron.

**Validation:** Nathan Moreau, Marco E. Mazevet, Anne-Laure Ejeil, Benjamin Salmon, Violaine Smaïl-Faugeron.

**Writing – original draft:** Hadrien Diakonoff, Sébastien Jungo, Nathan Moreau, Marco E. Mazevet, Violaine Smaïl-Faugeron.

**Writing – review & editing:** Benjamin Salmon.

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
