## [Decision Letter · Decision Letter 0]

12 Nov 2021

PONE-D-21-29169Application of recommended preventive measures against COVID-19 could help mitigate the risk of SARS-CoV-2 infection during dental practice: results from a follow-up survey of French dentistsPLOS ONE

Dear Dr. Smail-Faugeron,

Thank you for submitting your manuscript to PLOS ONE. After careful consideration, we feel that it has merit but does not fully meet PLOS ONE’s publication criteria as it currently stands. Therefore, we invite you to submit a revised version of the manuscript that addresses the points raised during the review process.

We look forward to receiving your revised manuscript.

Kind regards,

Sanjay Kumar Singh Patel, Ph.D.

Academic Editor

PLOS ONE

Reviewers' comments:

Reviewer's Responses to Questions

**Comments to the Author**

1. Is the manuscript technically sound, and do the data support the conclusions?

Reviewer #1: Yes

Reviewer #2: Yes

2. Has the statistical analysis been performed appropriately and rigorously? 

Reviewer #1: Yes

Reviewer #2: Yes

3. Have the authors made all data underlying the findings in their manuscript fully available?

Reviewer #1: Yes

Reviewer #2: Yes

4. Is the manuscript presented in an intelligible fashion and written in standard English?

Reviewer #1: Yes

Reviewer #2: Yes

5. Review Comments to the Author

Reviewer #1: The increased social contact is the main reason for the spread of COVID-19 and dentists are at high risk of infection due to their social interaction during the pandemic. This study investigates the French dentist population between July and September 2020 to examine the risk associated with COVID-19. An online questionnaire with 32 questions was divided into eight sections to collect the data about the risk associated with COVID-19, and then the data was subjected to proper statistical analysis. The present manuscript is well written and easy to understand. However, there are few formatting errors that could be resolve in the final submission. Therefore, there is technically no deficiency in the manuscript for rejection. The manuscript discusses few limitations, such as the prevalence of COVID-19 among dentists has been underestimated as only one-third of respondents have been tested, and second, how it is not possible to establish a causal relationship between being tested COVID-19 positive wearing a surgical mask during non-aerosol generating procedures. It is heartening to see authors discussing their results and its limitation in the same space. This study is relevant to publish in the current COVID-19 pandemic as it will help the clinician to understand future COVID-19 waves or another pandemic.

Reviewer #2: The manuscript entitled “Application of recommended preventive measures against COVID-19 could help mitigate the risk of SARS-CoV-2 infection during dental practice: results from a followup survey of French dentists” by Diakonoff et al., explored whether implementation of preventive measures, including adequate personal protective equipment (PPE) and room aeration between patients has had an impact on the contamination of dentists. Authors have concluded that wearing surgical masks during non-aerosol generating procedures was a risk factor of COVID-19, whereas reducing the number of patients was a protective factor. This is an interesting study; however, the manuscript still needs some changes.

Suggestions:

1. It will be interesting to know the COVID-19 risk among other health worker other than dentist during the same period.

2. At least one additional Figure (illustration) may be provided as to highlight the summary or prospect of this study.

3. Figures quality may be improved (high resolution).

---

## [Author Response · Author response to Decision Letter 0]

30 Nov 2021

Dear Prof. Joerg Heber,

Dear Editors,

Thank you for giving us the opportunity to improve our manuscript and resubmit our paper for further consideration by PLOS ONE. We thank the reviewer for his/her comments and suggestions. Each of the comments and/or issues raised by the reviewer has been thoroughly addressed, detailed in a point-by-point response hereafter.

Please find enclosed the revised version of our manuscript incorporating all of the changes. All authors have read and approved the revised version of the manuscript; the manuscript has not been published and is not being considered for publication elsewhere, in whole or in part, in any language,

Thank you for further considering our manuscript, which we hope is now suitable for publication in PLOS ONE.

Sincerely yours,

Dr Violaine Smaïl-Faugeron

Université de Paris – AP-HP

Paris, France

Journal requirements

RESPONSE: The manuscript has been modified accordingly, including the file naming. 

RESPONSE: In adherence to French bylaw (Jardé law) requirements applicable at time of writing, data collection was conducted under strict control of the French data regulation committee (CNIL). Participation was voluntary, anonymous, and non-incentivized, and participants were informed of the data collection prior to the beginning of the survey. As such, participating in the survey was construed as implicit consent.

Revised Methods section, page 4: “Participants were informed of the data collection, study aims and relevant data protection measures.” 

RESPONSE: We added a new reference: “Colomb-Cotinat M, Poujol I, Monluc S, Vaux S, Olivier C, Le Vu S, et al. Burden of COVID-19 on workers in hospital settings: The French situation during the first wave of the pandemic. Infectious diseases now. 2021;51(6):560-3.”

Review Comments to the Author

1. It will be interesting to know the COVID-19 risk among other health worker other than dentist during the same period.

RESPONSE: We thank the reviewer for his/her comment. We added a subsection in Discussion section to compare the COVID-19 risk between other health workers and dentists. 

 Revised Discussion section, page 11: “At the time of data collection (September 8, 2020), our results confirmed that there was no strong evidence to confirm that dentists were at higher risk of COVID-19 than the general population (3.6% of dentists vs. 5.2% of the general population, www.santepubliquefrance.fr), workers in hospital settings (3.4%), nor than healthcare workers (4.0%) [13].”

2. At least one additional Figure (illustration) may be provided as to highlight the summary or prospect of this study.

RESPONSE: We thank the reviewer for his/her comment. We added an additional Figure to highlight the prospect of this study. See Graphical abstract. 

3. Figures quality may be improved (high resolution).

RESPONSE: We thank the reviewer for his/her comment. We have improved figures quality.

---

## [Editor Report · Decision Letter 1]

2 Dec 2021

Application of recommended preventive measures against COVID-19 could help mitigate the risk of SARS-CoV-2 infection during dental practice: results from a follow-up survey of French dentists

PONE-D-21-29169R1

Dear Dr. Smail-Faugeron,

We’re pleased to inform you that your manuscript has been judged scientifically suitable for publication and will be formally accepted for publication once it meets all outstanding technical requirements.

Kind regards,

Sanjay Kumar Singh Patel, Ph.D.

Academic Editor

PLOS ONE

---

## [Editor Report · Acceptance letter]

7 Dec 2021

PONE-D-21-29169R1 

Application of recommended preventive measures against COVID-19 could help mitigate the risk of SARS-CoV-2 infection during dental practice: *results from a follow-up survey of French dentists*

Dear Dr. Smail-Faugeron:

I'm pleased to inform you that your manuscript has been deemed suitable for publication in PLOS ONE. Congratulations! Your manuscript is now with our production department. 

Kind regards, 

on behalf of

Dr. Sanjay Kumar Singh Patel 

Academic Editor

PLOS ONE